# Inoculation of Indigenous Arbuscular Mycorrhizal Fungi as a Strategy for the Recovery of Long-Term Heavy Metal-Contaminated Soils in a Mine-Spill Area

**DOI:** 10.3390/jof9010056

**Published:** 2022-12-29

**Authors:** Gloria Andrea Silva-Castro, Custodia Cano, Silvia Moreno-Morillas, Alberto Bago, Inmaculada García-Romera

**Affiliations:** Departamento de Microbiología del Suelo y Sistemas Simbióticos, Estación Experimental del Zaidín, Consejo Superior de Investigaciones Científicas (EEZ-CSIC), C/Profesor Albareda, 1, 18008 Granada, Spain

**Keywords:** arbuscular mycorrhizas, heavy metals, indigenous fungi, phytoremediation, metal tolerance, soil pollution

## Abstract

Symbiotic associations with arbuscular mycorrhizal fungi (AMF) offer an effective indirect mechanism to reduce heavy metal (HM) stress; however, it is still not clear which AMF species are more efficient as bioremediating agents. We selected different species of AMF: *Rhizoglomus custos* (Custos); *Rhizoglomus* sp. (Aznalcollar); and *Rhizophagus irregularis* (Intraradices), in order to study their inoculation in wheat grown in two soils contaminated with two levels of HMs; we tested the phytoprotection potential of the different AMF symbioses, as well as the physiological responses of the plants to HM stress. Plants inoculated with indigenous Aznalcollar fungus exhibited higher levels of accumulation, mainly in the shoots of most of the HM analyzed in heavily contaminated soil. However, the plants inoculated with the non-indigenous Custos and Intraradices showed depletion of some of the HM. In the less-contaminated soil, the Custos and Intraradices fungi exhibited the greatest bioaccumulation capacity. Interestingly, soil enzymatic activity and the enzymatic antioxidant systems of the plant increased in all AMF treatments tested in the soils with both degrees of contamination. Our results highlight the different AMF strategies with similar effectiveness, whereby Aznalcollar improves phytoremediation, while both Custos and Intraradices enhance the bioprotection of wheat in HM-contaminated environments.

## 1. Introduction

The mining industry is not only a leading factor in economic development in various countries but also an significant source of metal pollution. One of the serious challenges worldwide is the contamination of soil with heavy metals because they are not biodegradable and can accumulate in living organisms and are transferred from one trophic level to another in the food chain [1,2]. Therefore, it is crucial to remove or at least reduce HM’s negative impacts from the contaminated soils on an urgent basis to ensure the safety of the environment and human health [3,4].

Physico-chemical soil remediation, through techniques such as soil washing, chemical oxidation, and electrochemical remediation, has significant disadvantages, including high costs, secondary pollution, and limited scope of application [5]. However, bioremediation, which can involve animal remediation, phytoremediation, and microbial remediation, provides a long-lasting, effective, and environmentally friendly method for the remediation of HM-polluted soil [6]. Phytoremediation takes advantage of certain plant characteristics to remove, contain, or stabilize HM in plant tissues, thus preventing dispersion and reducing the impact of these contaminants on the environment [7]. Nevertheless, the effectiveness of phytoremediation is limited due to the chemical forms of metals, as well as the concentration, spatial distribution, solubility, and bioavailability of contaminants in the soil. 

Most plants have a certain degree of toxicity at high concentrations of metals and metalloids [8]. As the slow growth of hyperaccumulators in HM-contaminated soils makes the phytoextraction process very slow, the use of symbiotic soil microorganisms has been proposed to improve the efficiency of phytoextraction [9]. Symbiotic associations with arbuscular mycorrhizal fungi (AMF), one of the predominant fungal root colonizers, offer an effective, indirect mechanism for reducing HM stress while, at the same time, improving the water and mineral nutritional status of the plant and promoting plant growth in a metal-stressed environment [10]. They also immobilize the contaminant at the root level, which is transported in smaller amounts to the shoots (phytostabilization) or efficiently translocated to the shoots (phytoextraction). These properties, when combined with several plant detoxification mechanisms, suggest that AMF could be considered beneficial for the functioning and restoration of HM-contaminated ecosystems [11,12]. The effectiveness of these interactions varies in relation to plants, AMF species, soil properties, the type of toxic metal, and the extent of metal toxicity [12].

Arbuscular mycorrhizae (AM) exhibit different tolerance levels depending on HM type and concentration. In particular, AMF species that belong to the genus Glomus have been used to increase the efficiency of phytoremediation practices employed in programs to rehabilitate certain areas [13]. In contrast, species from other genera, such as Acaulospora and Gigaspora, have been less studied for this purpose [14]. AM symbiosis is known to increase plant tolerance to various abiotic stresses, such as cold, drought, salinity, heavy metal, and emerging pollutants, by decreasing reactive oxygen species (ROS) accumulation; increasing the activities of antioxidant enzymes, including superoxide dismutase (SOD), catalase (CAT), and peroxidase (POD); by improving photosynthesis; and by promoting nutrient uptake [15].

The aims of the research undertaken are to assess the phytoremediation potential of different plant–microbe symbionts and to analyze the physiological responses of plants to HM stress.

## 2. Materials and Methods

### 2.1. Soil

The soils studied are located in the upper sector of the Green Corridor of the Guadiamar River [16]. Given their high sand content (>50%), these soils are classified as having a sandy loam texture. Soil 1 showed limited vegetation growth, suggesting that it is still contaminated by heavy metals (HMs); in contrast, soil 2 is considered a recovered soil because it harbors vegetation native to the area. This was confirmed by HM analyses carried out on both soils (Table 1 and Table 2). In each of the corresponding areas, soil samples were randomly taken from the upper soil layer (0–10 cm), with three replicates composed of 5 soil sub-samples (center and four corners of a square meter), and intensively homogenized to provide a single sample for each sampling point.

### 2.2. AMF Inocula

The mycorrhizal inocula used were the *Rhizoglomus* sp. fungus (Aznalcóllar MUCL47213), a strain owned by Cusal Ingenio S.L., which was isolated a few years after the Aznalcóllar disaster on an island of surviving vegetation in the Guadiamar Green Corridor; *Rhizophagus irregularis* DAOM 197198 [17], formerly known as *Glomus intraradices* (Intraradices); and *Rhizoglomus custos* MUCL47214 (Custos), originally isolated on the banks of the Riotinto river near Nerva in Huelva (Spain) [18]. AM fungal inoculation was carried out by adding a 1 cm^3^ cube of AM-in vitro issued inoculum [19] containing high amounts of AM propagules (spores, active extraradical hyphae, and aseptic root pieces colonized by AM intraradical mycelium) of the different AM used. AM inoculation was performed during seedling transplanting, and special care was taken for fungal propagules to be in close contact with roots, thus inducing quick colonization.

### 2.3. Experimental Design

The experiment was set up in a series of identical polypropylene pots with an individual total volume of 0.3 L. Approximately 300 g of metal-contaminated soil was placed in each pot. The experimental design consisted of a random factorial system with two variation factors. The first factor refered to the level of soil contamination, while the second factor related to the uninoculated soil and the soil inoculated with the three AMF. Five repetitions were established for each treatment. A 10-day-old wheat plant (*Triticum aestivum L.*) was planted in each pot. The experiment was carried out under greenhouse conditions (supplementary light 25/19 °C and 50% relative humidity), and the plants were watered regularly to maintain humid conditions. After 45 days, the soil samples from each pot were homogenized, sieved (2 mm mesh), and subdivided into three subsamples. The first subsamples were used for the determination of humidity, the second for the analysis of soil enzymatic activities, and the third for the determination of the HM concentrations. Likewise, wheat plants were harvested, divided into roots and shoots, analyzed separately, and then divided into different subsamples (three for roots and four for shoots) to determine plant biomass, percentage of mycorrhization, HM content, and enzymatic activities.

### 2.4. Soil Analysis

The physico-chemical characteristics of each soil sample were determined according to standard methods [20] after the soil samples were dried and sieved (2 mm). Soil pH and electrical conductivity (EC) were measured in soil:water suspensions of 1:2.5 and 1:5, respectively. The percentage of nitrogen and carbon was also determined.

The total concentrations of various HMs (Pb, Zn, Cu, Cr, Ni, As, Co, and Cd) and other elements (P, Fe, Mn, Ca, and K), expressed in mg kg^−^^1^, were determined by an X-ray fluorescence analyzer (XRF), NITON GOLD XL3T-950 (Thermo Fisher Scientific, Tewksbury, MA, USA). Precision and accuracy were calculated by measuring a certified reference material (CRM052–050 RT-Corporation Limited, Salisbury, UK).

To determine β-glucosidase activity, the method described by [21] was followed. Thus, 1 g of dry soil was mixed with 5 mL of a p-nitrophenyl β-D-glucopyranoside solution dissolved at a concentration of 0.025 M in MUB buffer (0.1 M, pH 6) and incubated at 37 °C for 2 h. The substrate used was transformed into p-nitrophenol by the action of β-glucosidase, and the concentration of this compound was determined at 400 nm after the addition of 1 mL of 0.5 M CaCl_2_ and 4 mL of 0.1 M THAM buffer pH 12. β-Glucosidase activity was expressed as μg p-nitrophenol g^−^^1^ h^−^^1^.

Dehydrogenase activity was determined according to the procedure described by [22]. Accordingly, 1 g of soil was incubated with 2 mL of substrate 2-p-iodophenyl-3-p-nitrophenyl-5-phenyltetrazolium (INT) 0.5% and 1.5 mL of 1 M Tris-HCl buffer pH 7.5 for 1 h at 40 °C. Next, the iodonitrotetrazolium formazan (INTF) formed was extracted with a 1:1 (*v*:*v*) mixture of ethanol and dimethyl-formamide, and its absorbance at 490 nm was measured. Dehydrogenase activity was expressed as μmol INTF g^−^^1^ h^−^^1^.

### 2.5. Plant Analysis

At day 45, after harvesting, the plants were dried in an oven at 70 °C for 24 h and used for the different determination. The first subsamples were used to record plant biomass and were expressed as root and shoot dry weight at the end of the experiment. 

A second set of samples was used to measure the concentration of HM in roots and shoots. Acid digestion was performed with 3 mL of nitric acid plus 3 mL of hydrogen peroxide in a Mars XP1500 Plus microwave for 1 h at 200 °C and 800 W. Subsequently, the digested samples were supplemented with distilled water to a final volume of 10 mL and then analyzed using an Avio^®^ 500 inductively coupled plasma-optical emission spectrometer (ICP-OES).

In addition, a third set of samples was used to determine the enzyme activities; these samples were frozen in liquid nitrogen after harvesting and maintained at −80 °C until tested. The enzymes were extracted with 50 mM Tris-HCl buffer (pH 7.8), containing 1% (*w*/*v*) albumin and 0.5% (*w*/*v*) cysteine. The extracts were centrifuged at 20,000× *g* for 20 min at 4 °C; the supernatants obtained were then used to determine enzymatic activity. The soluble proteins were determined using Bradford reagent (Bio-Rad) [23] and BSA as standard. 

Glutathione reductase (GR; EC 1.6.4.2) activity was assayed by the decrease in absorbance at 340 nm due to the conjugation of oxidase glutathione (GSH) with nicotinamide adenine dinucleotide phosphate (NADPH) (ɛ = 6.22 mM^−1^cm^−1^). The specific activity was expressed as nmol min^−1^ mg^−1^ of protein. Ascorbate peroxidase (APX EC 1.11.1.11) activity was determined by monitoring the H_2_O_2_-dependent oxidation of ascorbate (ASC) at 290 nm in a reaction mixture containing 0.1 M Tris-acetate buffer (pH 6.4), 350 μM ASC, 170 μM H_2_O_2_, and 50–100 μg protein. The extinction coefficient was 2.75 mM^−1^cm^−1^ [24]. Catalase (CAT; EC 1.11.1.6) activity was determined according to the method described by [25] by measuring the decomposition of H_2_O_2_ at 240 nm (ɛ = 39.58 × 10^−3^ M^−1^ cm^−1^). The specific activity was expressed as nmol min^−1^ mg^−1^ of protein. Total superoxide dismutase (SOD; EC 1.15.1.1) activity was defined as the amount of enzyme that causes 50% reduction in cytochrome c by superoxide radicals according to the technique described by [26]. The specific activity was expressed as U mg^−1^ of protein.

The fourth set of subsamples of roots was used to measure the percentage of mycorrhization according to the staining methodology described by [27], and colonization was quantified using the method devised by [28].

### 2.6. Statistical Analysis

Mean values and respective standard deviations of the parameters were calculated for each set of samples. The data were analyzed by one-way analysis of variance (ANOVA) and post-hoc analysis using Duncan tests with *p* < 0.05 as statistical significance to evaluate the differences in the impact of the application of the three AMF inoculated in each of the soils. Ordination analysis (PCA) was used to determine the relationships between the AMF inoculated (Aznalcollar, Intraradices, and Custos), enzymatic activity in soil (dehydrogenase and β-glucosidase), the percentage of mycorrhization in plant biomass, and plant enzymatic activities, as well as the concentrations of HM detected in roots and shoots [29].

## 3. Results

### 3.1. Changes in Soil

#### 3.1.1. Physico-Chemical Characteristics

The soils studied are located in the upper sector of the Green Corridor of the Guadiamar River. Given their high sand content (>50%), these soils are classified as having a sandy loam texture. Soil 1 showed no vegetation, suggesting that it was still contaminated by heavy metals (HMs); in contrast, soil 2 is considered a recovered soil because it harbored vegetation native to the area. This was confirmed by HM analyses carried out on both soils (Table 1 and Table 2). 

Soil 1 presented an As concentration six-fold higher than that for soil 2, a four-fold higher concentration of Pb, and a two-fold higher concentration of Cu. No differences in chemical concentrations were found due to mycorrhizal activity, except for a decrease in Ni mediated by Custos inoculation and an increase in Cr mediated by Intraradices inoculation, both of which were observed in soil 2.

The most important differences exhibited by the two soils before and after AMF inoculation are with regard to pH, which sharply increased in soil 2 (4.24–4.28 to 7.69–7.74); electrical conductivity, which decreased from 3.36–3.56 to 0.84–0.87; and a decrease in P concentrations (Figure 1). In this latter case, the decrease depends on the AMF inoculated, with the most significant reduction being observed with Aznalcollar (6.15 mg lower P in soil 1, 13.76 mg lower P in soil 2) and Custos (5.1 mg lower P in soil 1). No significant differences were found between soils or treatments for N or C and the other elements analyzed (Fe, Mn, Ca, and K).

#### 3.1.2. Enzymatic Activities

Dehydrogenase and β-glucosidase enzymatic activities measured at the end of the experiment were used as general indicators of the biological activity of soils [30] (Figure 2). 

Both showed higher values in soil 2 as compared to soil 1, probably due to the physico-chemical conditions such as pH, conductivity, and the concentrations of HM (Table 1). In soil 1, Aznalcollar and Custos inoculation increased dehydrogenase activity, which differed significantly from that in uninoculated soil (Figure 2A); however, these AMF shown a decrease in dehydrogenase activity in soil 2. In the case of β-glucosidase activity (Figure 2B), again for soil 1, a significant increase was observed for all three AMF treatments with respect to the control treatment, while a decrease in this activity was observed in soil 2.

### 3.2. Plant Analysis

#### 3.2.1. Biomass

Although the shoot dry weight in soil 1 did not differ significantly between treatments, Custos inoculation significantly increased root dry weight when compared to the control (Figure 3A). Regarding soil 2 (Figure 3B), plants inoculated with all three AMFs showed higher dry weight in both shoots and roots with respect to the non-inoculated plants, although, in the case of Aznalcollar, these differences were not significant.

#### 3.2.2. AMF Colonization

AM colonization was observed in the roots of all the plants studied in both soils, as shown in Figure 4. In general, the roots from soil 2 presented a higher rate of AMF colonization as compared to soil 1. Inoculation of the different AMFs promoted a clear and significant increase in root colonization in soil 1 (from 5% in uninoculated soil to around 15% in AMF treatments). However, no significant differences were found between AMF treatments. In soil 2, a significant increase of over 20% in mycorrhization was observed in inoculated plants, mainly due to the Aznalcollar and Intraradices treatments.

#### 3.2.3. Heavy Metal Accumulation

Heavy metal analyses of plant material collected at the end of the experiment are shown in Table 3. Significant values showing HM accumulation with respect to control plants are shown in red, while the significant values showing HM depletion with respect to control plants are shown in blue. 

At first glance, it can be seen that, in soil 1, the data in red (accumulation) are concentrated in the Aznalcollar treatment, while the data in blue (depletion) are localized in the Intraradices and Custos treatments. More precisely, Aznalcollar inoculation promoted a higher level of HM accumulation in shoots, with significant differences for Cu, Zn, Co, As, Cr, and Pb being observed with respect to S1. No significant differences were detected for most of the elements analyzed in the depletion promoted by Custos and Intraradices. However, Intraradices showed a significant decrease in the concentrations of Zn and Pb in roots and of Co in shoots. In the case of Custos, depletion was only detected with regard to Zn in roots and Co in shoots.

In the case of soil 2, at first glance, the data in red predominated, indicating that bioaccumulation was prevalent. Custos mainly increases the bioaccumulation of As, Cr, and Pb in shoots; Intraradices significantly increases Cu, Cr, and Pb accumulation in roots; and Aznalcollar’s HM content significantly differs from the control, mainly with regard to Cr in roots and partially for Zn and Ni in shoots.

#### 3.2.4. Enzymatic Antioxidant Systems

The presence of HM induces the production of reactive oxygen species (ROS) in cells, which causes cellular damage at the membrane level. Plants have mechanisms, such as the enzymatic antioxidant system, capable of controlling the massive levels of ROS production. Enzymatic glutathione reductase (GR), ascorbate peroxidase (APX), catalase (CAT), and superoxide dismutase (SOD) activity in roots and shoots, with and without the AMF selected, was determined, and the results are shown in Figure 5.

The levels of GR in roots did not differ significantly between the control and mycorrhizal plants in the two soils studied. However, GR activity increased in the shoots of the mycorrhizal plants in both soils and showed significant differences with respect to the control plants in both soils and between treatments only in soil 2 (Figure 5A,B). Regarding APX activity, no significant increases were detected in mycorrhizal roots in soil 1 with respect to the control treatment. This was not the case for shoots, where Aznalcollar- and Intraradices-treated plants showed higher levels of APX activity. With respect to soil 2, this enzyme was only detected in Aznalcollar and Intraradices roots (Figure 5C,D). CAT activity in soil 1 wheat plants increased in Aznalcollar roots, while shoots did not show any significant differences. 

In soil 2, Intraradices treatment showed a significant increase in root CAT activity, while such significant differences were evident in Aznalcollar- and Intraradices-treated shoots (Figure 5E,F). SOD analysis revealed higher activity in soil 1 roots for all three mycorrhizal treatments, as well as in Intraradices- and Custos-treated shoots. With regard to soil 2, this enzyme was only detected in roots after inoculation with Aznalcollar and in shoots with Intraradices, with a level lower than the control being observed (Figure 5G,H).

#### 3.2.5. Principal Component Analysis Test

The combined effect of AMF inoculation on HM concentrations, antioxidant enzyme activities, and soil biological responses was analyzed using the Principal Component Analysis (PCA) test [29], with the results obtained shown in Figure 6A (soil 1) and Figure 6B (soil 2). The upper right quadrant of the PCA graph for soil 1 (Figure 6A) shows a clear association of the Aznalcollar inoculation with accumulations of most of the HMs analyzed; this association is also linked to the CAT enzymes in roots and the GR and APX enzymes in shoots. However, Intraradices and Custos do not increase the accumulation of HMs but rather has a bioprotective effect on the plant, which generates the production of GR in roots and of SOD and CAT in shoots.

In soil 2, Aznalcollar inoculation induces concentrations of Zn and Ni in the shoot and of Cr in roots and is also positively related to the production of SOD enzymes in roots and of CAT in shoots (Figure 6B). On the other hand, Intraradices and Custos are located in the lower left quadrant, suggesting that they both present a similar behavior with regard to the accumulation of metals such as As, Pb, and Cr in shoots and Cu and Zn in roots. It can be seen that this accumulation process is linked to the high level of mycorrhization and the production of GR in shoots.

## 4. Discussion

Arbuscular mycorrhizal (AM) plants are known to exhibit an extra tolerance to many types of biotic and abiotic stresses [12,31,32,33]. Among the latter, HM contamination has been widely studied, and the results obtained have pointed to AM inoculation as an important, sustainable strategy that enhances the capacity of plants to remove and/or sequestrate contaminants from the soil [14,34,35]. In this study, the potential of AM inoculation to restore HM-contaminated soil in a mine-spill area of high ecological value is tested. To do this, three different AM fungal strains were used: *Rhizophagus irregularis*, formerly known as *Glomus intraradices*, abbreviated to Intraradices, a model AM fungus, as an internal control; *Rhizoglomus custos*, formerly known as *Glomus custos* or *Rhizophagus custos*, abbreviated to Custos, a fungus isolated from a naturally HM contaminated soil, which has already shown its capacity to confer extra HM endurance on plants [18,36]; and an indigenous AMF strain, initially identified as *Rhizophagus irregularis*, abbreviated to Aznalcollar, which was isolated from the same HM-contaminated area under study some years after the mine-spill accident.

One of the recognized benefits of AMF is their ability to improve plant nutritional status, thereby promoting plant growth and tolerance to stress conditions [10,35]. The restoration of HM-contaminated soils may be due to the greater absorption of these elements in the mycorrhizosphere [35]; it is therefore important to study the impact of AMF inoculation on soil biology and chemical properties. AMF are typically known to improve plant growth and dry weight and to protect against stress caused by metals and metalloids [8] by favoring increased nutrient uptake, especially phosphorus (P), which is involved in plant detoxification mechanisms [7]. In our study, inoculation with AMF showed a decrease in the P concentrations in the soils studied. Aznalcollar was more efficient in capturing P in both soils (S1 and S2) and Custos mainly in the soil with higher stress levels due to HM contamination (S1) and a pH of under 5. This suggests that these AMF assist in the transport of these nutrients to the plant, while favoring its nutritional conditions and consequently its development [37]. This is in line with the studies conducted by [38] whose authors observed that the absorption of phosphorus in tomato plants changed according to the different species of inoculated mycorrhizae.

The different P absorption rates of the various mycorrhizal fungi used to inoculate wheat plants are reflected in the plant’s dry weight. The higher root biomass of plants inoculated with Custos in S1 than the control could be directly related to the higher uptake of P from the soil. In the case of S2, the plant inoculated with Intraradices and Custos showed a higher dry weight than the control, although the amount of P absorbed from the soil was not the highest. However, as some researchers have reported, AMF have other mechanisms to increase plant growth and to withstand the negative effects of HM accumulation [10,12,39]. Numerous experiments have shown that the plant growth response to AMF is not always positive and depends on the plant and fungus genotypes, as well as on environmental conditions [40]. When the interaction between the plant and the fungus is not reflected in plant growth, there are other mechanisms at play that modify the plant’s defense systems that affect its physiology and improve the response to biotic and abiotic stresses [41]. These results indicate the necessity of including AMF inoculation in protocols for the restoration of vegetal cover in soils affected by HM contamination.

Enzymatic activity is considered as an indicator to evaluate the impact of treatments on the biological conditions of the soil [15]. As the β-glucosidase and dehydrogenase enzymatic activities were impacted differentially by the level of pollution, the less-contaminated soil (S2) showed higher levels of dehydrogenase and β-glucosidase than the more contaminated soil (S1). Because dehydrogenase and β-glucosidase are indicators of the microbial redox system and the carbon cycle, respectively, low values for these activities are expected in the mining soil [42]. According to [43] a possible explanation for this enzymatic inactivity is protein denaturation through the replacement of active groups of the enzyme by HM. AMF inoculation produced different effects on soil enzymatic activities depending on the degree of soil contamination. Thus, in the slightly contaminated soil (S2), the AMF tested did not increase soil enzymatic activities, while these fungi, mainly Aznalcollar and Custos, in highly contaminated soils (S1) managed to increase dehydrogenase and glucosidase activity. These results suggest that Aznalcollar and Intraradices indirectly stimulate biological activity by modifying soil microbial communities. According to [44], the soluble substances released by the extraradical mycelium of *Glomus intraradices* have both stimulatory and antagonistic effects on the individual growth of fungi and bacteria. In addition, some authors have suggested that the exudates produced by the AM mycelium can play an important role in the interactions of the different microbial communities in the soil, which are involved in nutrient cycles, their biological control, and their bioremediation [45].

A first issue that arises is the real potential of externally-inoculated AMF being established in natural areas where other AMF are already colonizing the existing vegetation. The results obtained for the percentage of mycorrhization show that this is indeed possible, and with great success: mycorrhizal colonization increases with all three AMF inoculants, reaching levels ranging from 12% (S1) to 25% (S2). Even though the colonization rate was low in our contaminated soil, it allows the plant to benefit from its fungal partner while maintaining integrity and self-control in roots [11,46]. In this context, it is important to note that mycorrhization levels of 80–90% are quite frequently reported in scientific studies and have, up to some time ago, been accepted as “normal” in order to obtain plant responses; however, at such levels, the root is almost “invaded” by the fungus, meaning that it may be so stressed that it influences results for which a physiological state of the root (and the plant) is required. It is also important to note that several studies have already demonstrated that exogenously inoculated AMF do not hamper natural colonization nor decrease either biodiversity or natural inoculum potential; on the contrary, a synergistic relationship between natural and exogenously introduced AMF populations has often been reported [47], if exogenous inoculations are performed under reasonable parameter conditions.

Although wheat is a plant species that tolerates various HMs, its extraction potential has been reported to be significantly increased by AMF synchronization [48,49,50]. Indeed, the results of this study show that AMF inoculation may become a key, natural tool to restore highly HM-contaminated soils. Moreover, our study strongly suggests that indigenous AMF behave better than non-indigenous AMF for this purpose, as Aznalcollar treatment has been shown to be the most effective in accumulating HMs such as Cu, As, Cr, and Pb, and, to a lesser extent, Zn and Co in the still heavily-contaminated S1 soil. This accumulation takes place mainly at the shoot level, which indicates that the use of this fungus would be compatible with bioextraction procedures for cleaning and restoring HM-contaminated soil. The symbiosis with Aznalcollar has the potential to increase phytoextraction efficiency, as has been reported previously with other fungal species [51]. These findings provide important information for developing the phytoremediation potential of metal-polluted arable soils via bioaugmented metal-accumulation provided by Aznalcollar fungi in wheat, as observed in previous studies [52]. In contrast, the strategy used by the non-indigenous AMFs, Custos, and Intraradices was to “exclude” HM, as though HM would be externally retained or somehow prevented from entering the plant. This indicates that these AMFs could be used to produce non-contaminated plants in soils whose HM content is excessive. Indeed, other studies have reported that the symbiotic association with AMFs offers different strategies for plants to reduce HM stress depending on the fungus, the plant, and the metal involved [11]. AMF may reduce the availability of metals and metalloids for plants by restricting their absorption and translocation in mycorrhizal plants [53,54,55].

Soil 2 presents much better physico-chemical conditions compared to soil 1, especially in terms of pH and electrical conductivity, which explains the presence of vegetation and indicate that the soil has recovered from the mine-spill disaster. The behavior of inoculated plants in this soil differs significantly from that of soil 1; Aznalcollar shows only high accumulation for Cr in roots and partially for Zn and Ni shoots, whereas the other two AMFs accumulate As, Cr, and Pb (Custos) and Cu, Cr, and Pb (Intraradices) in soil 2. These differences between the two soils highlight the importance of selecting the appropriate AMF to fight against a specific condition in soils. In our study, Aznalcollar is a “rough” indigenous fungus, which could be used to deal with the worst possible HM conditions. In this respect, Ref. [46] have suggested that HM-contaminated arbuscular fungi naturally existing in soils, which develop mechanisms similar to those of accumulator plants, are more tolerant to these elements. Once HM conditions improve, other AMFs could share these tasks. Thus, Custos and Intraradices would be “second row fighters”, which, like most AMF tested up to now, also show HM remediation potential, although to a lesser extent than indigenous AMF.

Besides HM accumulation or exclusion, other strategies are used by plants to endure HM contamination. Enzymatic antioxidant systems play a pivotal role in ROS quenching, which may be the perfect combination to make plant HM tolerance more effective. AMF inoculation is known to ameliorate stress by enhancing antioxidant enzyme activity [56]. The increase in antioxidant activity in AMF-inoculated plants can be attributed to the role of mycorrhizal hyphae in Zn and Cu transport, which act as co-factors of these enzymes [57]. In the presence of HM, plants with different AMFs have been shown to employ antioxidant enzyme systems, which can protect the structure and function of the membrane system and maintain the redox state of cells, as defense mechanisms against stress caused by the metal [58,59,60,61]. SOD dismutates superoxide radicals to H_2_O_2_, which is then converted into H_2_O and O_2_ by CAT and APX, while GR sustains the reduced status of GSH via the ascorbate–glutathione pathway and plays an essential role in maintaining the sulfhydryl (–SH) group [62]. Interestingly, glutathione reductase (GR) activity greatly increased in all AMF treatments tested as compared to control plants: more than five-fold in shoots in both S1 and S2. A similar response was obtained for ascorbate peroxidase (APX), which increased sharply in Aznalcollar- and Intraradices-inoculated shoots (soil 1) and roots (soil 2). In heavily contaminated soil 1, though undetectable in both uninoculated roots and shoots, SOD activity was quite high in all three AMF treatments. CAT activity increased in Aznalcollar-inoculated roots in S1, while, in S2, this activity increased in Aznalcollar-treated shoots and Intraradices-inoculated shoots and roots. Generally, higher antioxidant enzyme activity could be due to enhanced ROS production under HM-induced oxidative stress, while a decrease in this activity suggests that ROS generation has exceeded the elimination capacity of the antioxidant enzymes, or that ROS may have inactivated the enzymes [5,63]. It is important to note that some of these enzymatic activities are quite cyclic and plastic and also depend on metal availability in the soil, the type of metal, and even the plant’s physiological state at a given moment [64]. Our results are, in fact, a reflection of a kind of “snapshot” of the plant’s physiology; nevertheless, AMF inoculation clearly promotes the activity of these enzymes, thus providing the plant with extra protection against HM damage through ROS emission.

In an overall picture of specific HM tolerance strategies, Principal Component Analysis indicates that Aznalcollar’s principal strategy against very heavy HM contamination (soil 1) involves bioaccumulation, mainly in shoots; conversely, the main strategy of Custos and Intraradices involves the promotion of enzymatic antioxidant activities. When HM conditions are less dramatic (soil 2), Custos and Intraradices show the greatest bioaccumulation capacity. It is important to understand the mechanisms at work in the different AMF isolates in order to select the most appropriate AMF in any given situation; more research is needed to dissect these mechanisms.

In summary, the inoculation of plants with AMF is a key strategy in making the recovery of HM-contaminated soil possible, particularly when indigenous fungi are selected. Hopefully, this will be a compulsory strategy in the not-so-distant future.

## Figures and Tables

**Figure 1 jof-09-00056-f001:**
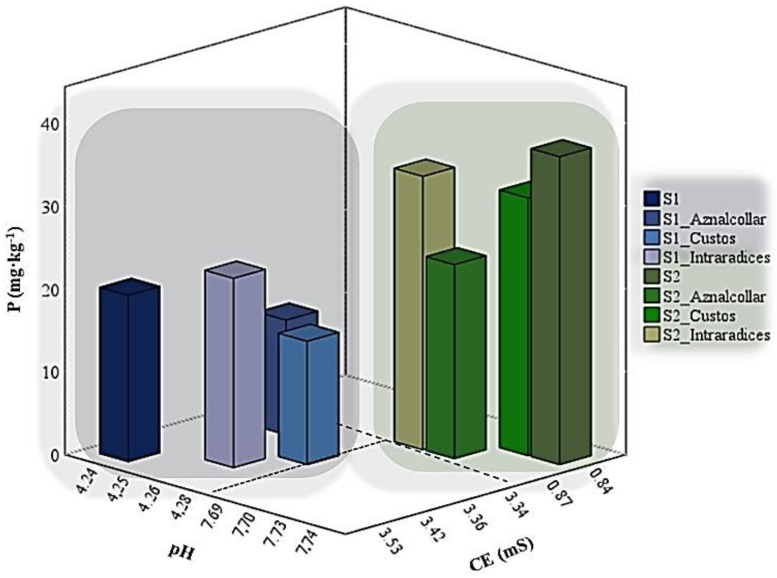
pH, electrical conductivity, and P concentrations in soils 1 (S1) and 2 (S2) inoculated with the three AMF selected (Aznalcollar, Intraradices, and Custos).

**Figure 2 jof-09-00056-f002:**
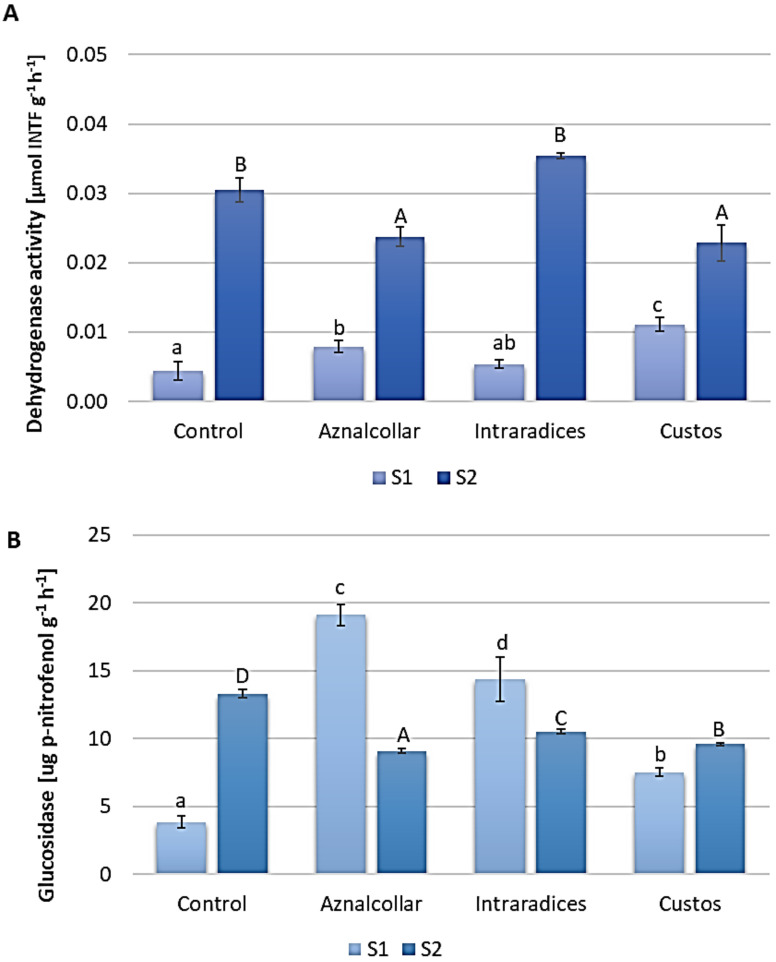
Enzymatic activity measured in soils 1 and 2 (S1 and S2) inoculated with the three AMF selected (Aznalcollar, Intraradices, and Custos): (**A**) dehydrogenase; (**B**) ß-glucosidase. Error bars represent one standard deviation from the mean (*n* = 3). Different letters indicate significant differences between treatments (*p* < 0.05) according to the Duncan post-hoc test; lowercase for S1 and capital letters for S2.

**Figure 3 jof-09-00056-f003:**
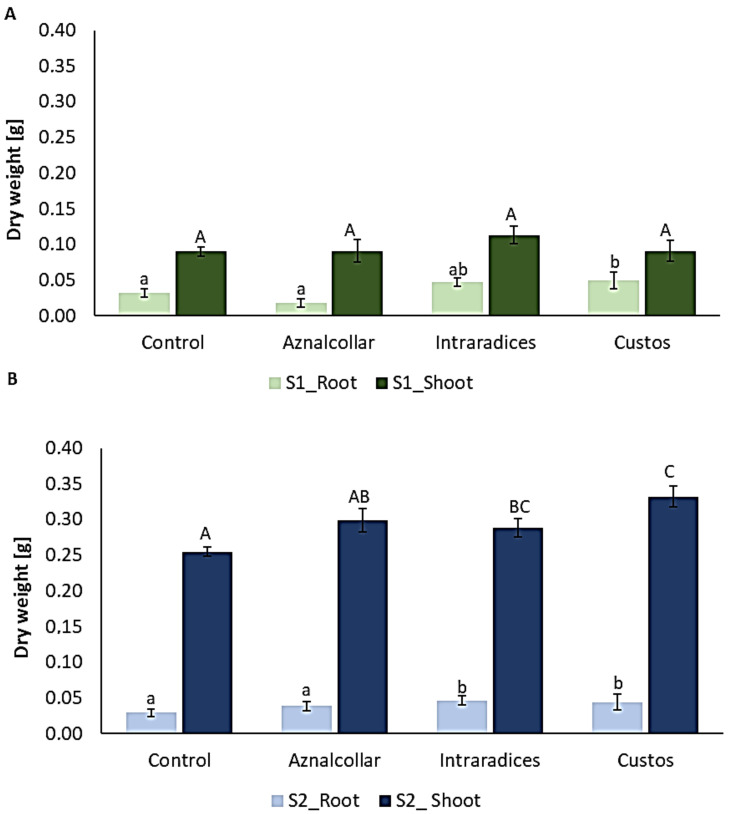
Total biomass of roots and shoots in soil 1 (S1) (**A**) and soil 2 (S2) (**B**) inoculated with the three AMF selected (Aznalcollar, Intraradices, and Custos). Error bars represent standard deviation. Different letters indicate significant differences between treatments (*p* < 0.05) according to the Duncan post-hoc test: lowercase for S1 and capital letters for S2.

**Figure 4 jof-09-00056-f004:**
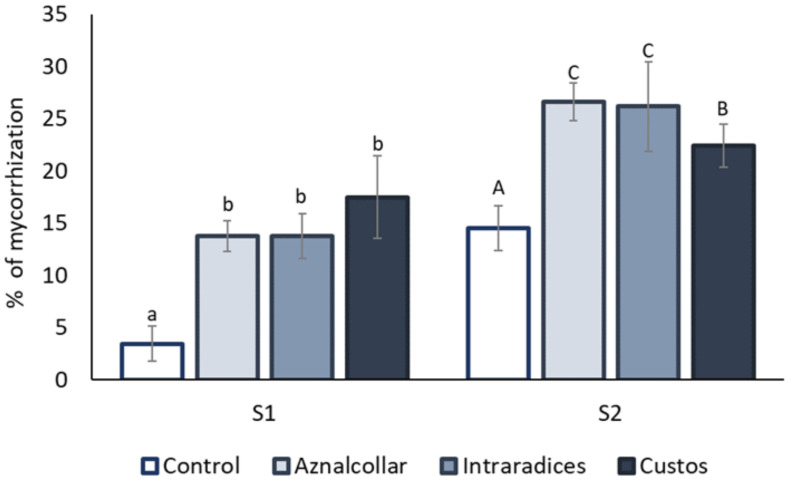
Percentages of root mycorrhization for the three AMF selected (Aznalcollar, Intraradices, and Custos). Values are means (*n* = 5). Different letters indicate significant differences between treatments (*p* < 0.05) according to the Duncan post-hoc test: lowercase for soil 1 (S1) and capital letters for soil 2 (S2).

**Figure 5 jof-09-00056-f005:**
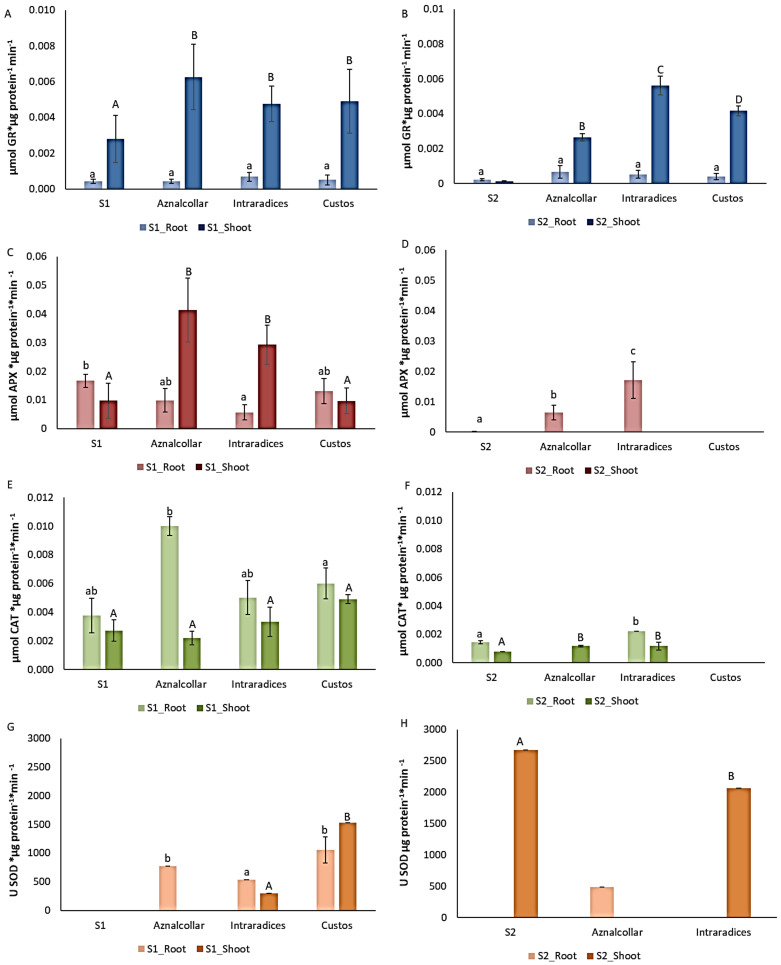
Enzymatic activities GR (**A**,**B**), APX (**C**,**D**), CAT (**E**,**F**) and SOD (**G**,**H**) in roots and shoots of control and mycorrhizal plants in soil 1 (S1, left column) and soil 2 (S2, right column). Error bars represent one standard deviation from the mean (*n* = 3). Different letters indicate significant differences between treatments (*p* < 0.05) according to Duncan post-hoc test: lowercase for S1 and capital letters for S2.

**Figure 6 jof-09-00056-f006:**
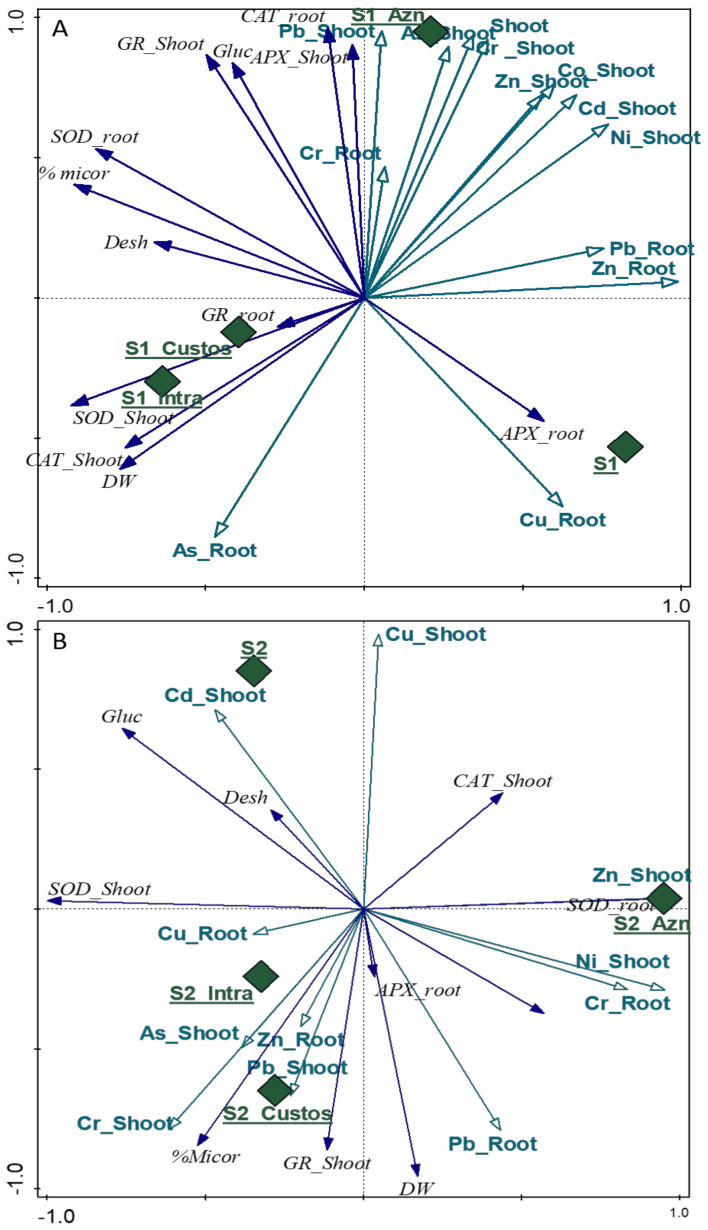
Principal component analysis (PCA) for the analyzed soil 1 (S1) (**A**) and soil 2 (S2) (**B**). The treatments used (control: S1 and S2; Aznalcollar: Azn; Intraradices: Intra; Custos: Custos) are correlated with green diamonds; the concentration of heavy metals in roots and shoots with green arrows; and the biological parameters (enzymatic antioxidant systems (APX, GR, CAT, SOD), enzymatic soil activity (Desh and Gluc), plant biomass (DW), and percentage of mycorrhization (%Micor) with dark blue arrows.

**Table 1 jof-09-00056-t001:** Chemical characteristics of soil 1 (Control) inoculated with the three selected AMF (Aznalcollar, Intraradices, and Custos).

	Soil 1			
	Control	Aznalcollar	Intraradices	Custos
Pb (mg kg^−1^)	395 ± 19 ^a^	396± 19 ^a^	394± 19 ^a^	395± 19 ^a^
Zn (mg kg^−1^)	238 ± 10 ^a^	235 ± 5 ^a^	238 ± 20 ^a^	238 ± 4 ^a^
Cu (mg kg^−1^)	121 ± 7.9 ^a^	124 ± 7.7 ^a^	125 ± 9.8 ^a^	125 ± 3.5 ^a^
Cr (mg kg^−1^)	82 ± 3.1 ^a^	87 ± 11 ^a^	74 ± 5.8 ^a^	87 ± 15 ^a^
Ni (mg kg^−1^)	38 ± 6.7 ^a^	39 ± 4.7 ^a^	38 ± 8.4 ^a^	40 ± 5.3 ^a^
As (mg kg^−1^)	242 ± 9.0 ^a^	240 ± 7.9 ^a^	239 ± 13 ^a^	249 ± 5.9 ^a^
Co (mg kg^−1^)	ND	ND	ND	ND
Cd (mg kg^−1^)	ND	ND	ND	ND

Different letters indicate significant differences between treatments (*p* < 0.05) according to Duncan post-hoc test. ND: not detected.

**Table 2 jof-09-00056-t002:** Chemical characteristics of soil 2 (Control) inoculated with the three selected AMF (Aznalcollar, Intraradices, and Custos).

	Soil 2			
	Control	Aznalcollar	Intraradices	Custos
Pb (mg kg^−1^)	96 ± 7.8 ^a^	95 ± 2.7 ^a^	96 ± 6.1 ^a^	94 ± 3.4 ^a^
Zn (mg kg^−1^)	237 ± 12 ^a^	247 ± 6.8 ^a^	244 ± 16 ^a^	237 ± 8.9 ^a^
Cu (mg kg^−1^)	61 ± 3.9 ^a^	63 ± 2.3 ^a^	62 ± 6.4 ^a^	63 ± 9.4 ^a^
Cr (mg kg^−1^)	77 ± 7.0 ^ab^	80 ± 3.9 ^bc^	85 ± 3.6 ^c^	74 ± 6.2 ^a^
Ni (mg kg^−1^)	33 ± 7.1 ^b^	33 ± 5.1 ^b^	30 ± 5.5 ^ab^	25 ± 4.3 ^a^
As (mg kg^−1^)	44 ± 3.5 ^a^	44 ± 1.3 ^a^	45 ± 2.3 ^a^	45 ± 4 ^a^
Co (mg kg^−1^)	ND	ND	ND	ND
Cd (mg kg^−1^)	ND	ND	ND	ND

Different letters indicate significant differences between treatments (*p* < 0.05) according to Duncan post-hoc test. ND: not detected.

**Table 3 jof-09-00056-t003:** Heavy metal concentration in roots and shoots in soil 1 (S1) and soil 2 (S2) inoculated with the three selected AMF (Aznalcollar, Intraradices, and Custos).

mg·kg^−1^	Treatments
S1	S1_Aznalcollar	S1_Intraradices	S1_Custos
**Cu ROOT**	103.66 ± 9.1 ^b^	78.13 ± 14.6 ^a^	81.17 ± 10.1 ^a^	87.28 ± 4.1 ^ab^
**Cu SHOOT**	36.27 ± 6.8 ^a^	81.16 ± 2.1 ^b^	31.51 ± 10.2 ^a^	22.57 ± 1.5 ^a^
**Zn ROOT**	308.71 ± 31.7 ^c^	258.13 ± 16.2 ^b^	180.16 ± 19.1 ^a^	175.05 ± 16.4 ^a^
**Zn SHOOT**	1273.15 ± 220.6 ^ab^	1894.77 ± 477.6 ^c^	637.17 ± 30.7 ^a^	944.58 ± 271.3 ^b^
**Co ROOT**	ND	ND	ND	ND
**Co SHOOT**	24.02 ± 8.7 ^b^	38.70 ± 9.1 ^c^	11.76 ± 6.3 ^a^	13.58 ± 5.2 ^a^
**As ROOT**	15.18 ± 8.6 ^a^	12.01 ± 5.5 ^a^	15.19 ± 1.1 ^a^	17.28 ± 11.5 ^a^
**As SHOOT**	4.75 ± 1.7 ^a^	10.40 ± 4.3 ^b^	6.63 ± 2.1 ^a^	2.77 ± 1.4 ^a^
**Cr ROOT**	17.35 ± 7.5 ^a^	23.06 ± 13.3 ^a^	26.63 ± 12.2 ^a^	12.17 ± 6.7 ^a^
**Cr SHOOT**	1.96 ± 0.8 ^a^	4.35 ± 1.6 ^b^	1.91 ± 1.1 ^a^	1.37 ± 0.7 ^a^
**Cd ROOT**	ND	ND	ND	ND
**Cd SHOOT**	4.15 ± 0.9 ^a^	5.38 ± 1.6 ^a^	2.89 ± 1.7 ^a^	2.87± 1.1 ^a^
**Pb ROOT**	64.08 ± 19.5 ^b^	60.28 ± 8.9 ^b^	28.94 ± 6.8 ^a^	47.94 ± 12.3 ^ab^
**Pb SHOOT**	5.98 ± 0.5 ^a^	22.40 ± 6.9 ^b^	13.40 ± 1.3 ^a^	5.42 ± 1.45 ^a^
**Ni ROOT**	ND	ND	ND	ND
**Ni SHOOT**	22.28 ± 8.6 ^ab^	27.87 ± 8.1 ^b^	12.04 ± 6.2 ^a^	11.48 ± 6.4 ^a^
	**S2**	**S2_Aznalcollar**	**S2_Intraradices**	**S2_Custos**
**Cu ROOT**	41.54 ± 7.4 ^a^	38.85 ± 5.4 ^a^	63.67 ± 5.8 ^b^	36.41 ± 3.8 ^a^
**Cu SHOOT**	43.0 ± 10.5 ^a^	36.60 ± 1.9 ^a^	32.65 ± 4.7 ^a^	31.54 ± 8.9 ^a^
**Zn ROOT**	73.79 ± 12.2 ^a^	76.41 ± 4.7 ^a^	85.71 ± 13.7 ^a^	76.02 ± 10.9 ^a^
**Zn SHOOT**	399.62 ± 73.1 ^a^	432.91 ± 46.8 ^b^	376.57 ± 54.3 ^a^	287.05 ± 27.8 ^a^
**Co ROOT**	ND	ND	ND	ND
**Co SHOOT**	ND	ND	ND	ND
**As ROOT**	ND	ND	ND	ND
**As SHOOT**	3.90 ± 1.2 ^a^	2.83 ± 0.7 ^a^	2.73 ± 0.9 ^a^	7.56 ± 4.0 ^b^
**Cr ROOT**	3.84 ± 1.2 ^a^	12.52 ± 1.1 ^b^	14.50 ± 1.9 ^b^	7.68 ± 2.9 ^ab^
**Cr SHOOT**	16.48 ± 7.0 ^a^	15.33 ± 6.6 ^a^	24.98 ± 3.8 ^b^	27.12 ± 7.8 ^b^
**Cd ROOT**	ND	ND	ND	ND
**Cd SHOOT**	0.54 ± 0.1 ^b^	0.48 ± 0.05 ^b^	0.19 ± 0.1 ^a^	0.31 ± 0.1 ^b^
**Pb ROOT**	4.59 ± 0.4 ^a^	4.48 ± 1.9 ^a^	15.04 ± 2.5 ^b^	12.69 ± 0.7 ^b^
**Pb SHOOT**	1.54 ± 0.4 ^a^	1.92 ± 0.0 ^a^	1.18 ± 0.3 ^a^	9.56 ± 0.1 ^b^
**Ni ROOT**	ND	ND	ND	ND
**Ni SHOOT**	7.26 ± 2.5 ^a^	11.99 ± 2.6 ^b^	8.02 ± 2.5 ^a^	9.01 ± 1.2 ^a^

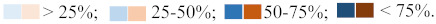
 Depletion (blue) and accumulation (brown) with respect to control. Different letters (a, b and ab) indicate significant differences between treatments (*p* < 0.05) according to the Duncan post-hoc test. ND: not detected.

## Data Availability

Not applicable.

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
