# Peer review of "Inoculation of Indigenous Arbuscular Mycorrhizal Fungi as a Strategy for the Recovery of Long-Term Heavy Metal-Contaminated Soils in a Mine-Spill Area"

_jof, 2022, doi:10.3390/jof9010056_

Round 1
Reviewer 1 Report
The present manuscript evaluated differents isolates of AM fungi associated with wheat in contaminated soils. The introduction is ok, and M&M is also well describing the differents methods used and plant growth conditions. About results and differences found among treatments specially plant growth and AM colonization in my view point were small. The Authors really believe that 30% of AM colonization is enough to protect plant health in the presence of HM ? The differences observed in plant growth in soil 1 and 2 are high, but in the same soil is too small, why ? The inocula of AMF was isolated from contaminated soil except G. intraradices, Is it true? If yes, plant growth promotion is almost the same, but the enzymes activity were different among treatment in soil 1 and soil 2, how the Authors explain it ? Figures needs to improve quality specialy Figure 1 and 2. Figure 2B Glucosidasas is in Spanish, please change. Figure 5 also need to improve quality.
Author Response
Thank you very much for the comments and suggestions, we have made all the suggested modifications and we attach the document of the answers point by point

Reviewer 2 Report
The manuscript studies the remediation of heavy metals by different AMF species, which is important for heavy metal contaminated soils.
The language was not clearly and needs major revision.
The introduction is not coherent and lacks logic. The research is aimed at heavy metal pollution in mining areas, but the introduction lacks this part.
The description of Materials and Methods was not clear. How to get soil 1 and soil 2? How to design the sampling point? Sampling method?
It is recommended that the basic information for soil 1 and soil 2 be presented separately.
Approximately 300 g of metal-contaminated soil was placed in each pot, how many AMF? Variety of wheat?
The information expressed in Table1 is not clear, and the letter marking is not correct. When Custos was inoculated, the Pb concentration was 395 ± 19 marked as a, while the Pb concentration of the control treatment was special 395 ± 19, but marked as b? The Zn concentration in the soil was higher after inoculation with Aznalcollar?
Figure 5 is recommended for high resolution.
Author Response

(The authors gave the same response as above.)

Reviewer 3 Report
1)why wheat was chosen for the experiment?
2_what cultivar was used?
3) Materials and Methods section line 120 ‘Dehydrohenase activity…’- should be written from a new line; the same for each parameter below
4)section 2.5- indicate that the analysis was achieved on dry samples
7)Table 1 the first column. Bold letters are not necessary
8) line 210 delete repetition ‘in soil 1’
9)Figure 3 use ‘g’ instead of ‘gr’
10) line 431 ‘[43] have suggested..’- use the authors names, that is ‘Basiru and Hijri [43] have suggested..’
11)Lines 465-473- add the title of the last section ‘Conclusions’
Author Response

(The authors gave the same response as above.)

Round 2
Reviewer 1 Report
The Authors made all changes that I suggested including the discussion about the results. I agree of the response, but I think that others factor probably are involved in the stress conditions in inoculated plants, the Authors did not consider the mycorrhizosphere effect and the microbial community in this microcosm at the contaminated soil and their influence in plant growth. I agree that low mycorrhization level could help plant in chemical stress conditions, but the microbial community must be also considered, but it is another chapter of the history. I suggest that in the future experiments the microbial community and mycorrhizosphere effect should be considered.